# Gradientless Descent: High-Dimensional Zeroth-Order Optimization

**Daniel Golovin, John Karro, Greg Kochanski, Chansoo Lee**
**Xingyou Song, Qiuyi (Richard) Zhang**[*]
Google Brain
`{dgg,karro,gpk,chansoo,xingyousong,qiuyiz}@google.com`

## Abstract

Zeroth-order optimization is the process of minimizing an objective $f(x)$, given oracle access to evaluations at adaptively chosen inputs $x$. In this paper, we present two simple yet powerful GradientLess Descent (GLD) algorithms that do not rely on an underlying gradient estimate and are numerically stable. We analyze our algorithm from a novel geometric perspective and present a novel analysis that shows convergence within an $\epsilon$-ball of the optimum in $O(kQ \log(n) \log(R/\epsilon))$ evaluations, for *any monotone transform* of a smooth and strongly convex objective with latent dimension $k < n$, where the input dimension is $n$, $R$ is the diameter of the input space and $Q$ is the condition number. Our rates are the first of its kind to be both 1) poly-logarithmically dependent on dimensionality and 2) invariant under monotone transformations. We further leverage our geometric perspective to show that our analysis is optimal. Both monotone invariance and its ability to utilize a low latent dimensionality are key to the empirical success of our algorithms, as demonstrated on BBOB and MuJoCo benchmarks.

## 1 Introduction

We consider the problem of zeroth-order optimization (also known as gradient-free optimization, or bandit optimization), where our goal is to minimize an objective function $f : \mathbb{R}^n \to \mathbb{R}$ with as few evaluations of $f(x)$ as possible. For many practical and interesting objective functions, gradients are difficult to compute and there is still a need for zeroth-order optimization in applications such as reinforcement learning (Mania et al., 2018; Salimans et al., 2017; Choromanski et al., 2018), attacking neural networks (Chen et al., 2017; Papernot et al., 2017), hyperparameter tuning of deep networks (Snoek et al., 2012), and network control (Liu et al., 2017).

The standard approach to zeroth-order optimization is, ironically, to estimate the gradients from function values and apply a first-order optimization algorithm (Flaxman et al., 2005). Nesterov & Spokoiny (2011) analyze this class of algorithms as gradient descent on a Gaussian smoothing of the objective and gives an accelerated $O(n\sqrt{Q} \log((LR^2 + F)/\epsilon))$ iteration complexity for an $L$-Lipschitz convex function with condition number $Q$ and $R = \|x_0 - x^*\|$ and $F = f(x_0) - f(x^*)$. They propose a two-point evaluation scheme that constructs gradient estimates from the difference between function values at two points that are close to each other. This scheme was extended by (Duchi et al., 2015) for stochastic settings, by (Ghadimi & Lan, 2013) for nonconvex settings, and by (Shamir, 2017) for non-smooth and non-Euclidean norm settings. Since then, first-order techniques such as variance reduction (Liu et al., 2018), conditional gradients (Balasubramanian & Ghadimi, 2018), and diagonal preconditioning (Mania et al., 2018) have been successfully adopted in this setting. This class of algorithms are also known as stochastic search, random search, or (natural) evolutionary strategies and have been augmented with a variety of heuristics, such as the popular CMA-ES (Auger & Hansen, 2005).

These algorithms, however, suffer from high variance due to non-robust local minima or highly non-smooth objectives, which are common in the fields of deep learning and reinforcement learn-

---

[*]Author list in alphabetical order.

ing. Mania et al. (2018) notes that gradient variance increases as training progresses due to higher variance in the objective functions, since often parameters must be tuned precisely to achieve reasonable models. Therefore, some attention has shifted into direct search algorithms that usually finds a descent direction $u$ and moves to $x + \delta u$, where the step size is not scaled by the function difference.

The first approaches for direct search were based on deterministic approaches with a positive spanning set and date back to the 1950s (Brooks, 1958). Only recently have theoretical bounds surfaced, with Gratton et al. (2015) giving an iteration complexity that is a large polynomial of $n$ and Dodangeh & Vicente (2016) giving an improved $O(n^2 L^2/\epsilon)$. Stochastic approaches tend to have better complexities: Stich et al. (2013) uses line search to give a $O(nQ \log(F/\epsilon))$ iteration complexity for convex functions with condition number $Q$ and most recently, Gorbunov et al. (2019) uses importance sampling to give a $O(n\bar{Q} \log(F/\epsilon))$ complexity for convex functions with average condition number $\bar{Q}$, assuming access to sampling probabilities. Stich et al. (2013) notes that direct search algorithms are invariant under monotone transforms of the objective, a property that might explain their robustness in high-variance settings.

In general, zeroth order optimization suffers an at least linear dependence on input dimension $n$ and recent works have tried to address this limitation when $n$ is large but $f(x)$ admits a low-dimensional structure. Some papers assume that $f(x)$ depends only on $k$ coordinates and Wang et al. (2017) applies Lasso to find the important set of coordinates, whereas Balasubramanian & Ghadimi (2018) simply change the step size to achieve an $O(k(\log(n)/\epsilon)^2)$ iteration complexity. Other papers assume more generally that $f(x) = g(\mathbf{P_A}x)$ only depends on a $k$-dimensional subspace given by the range of $\mathbf{P_A}$ and Djolonga et al. (2013) apply low-rank approximation to find the low-dimensional subspace while Wang et al. (2013) use random embeddings. Hazan et al. (2017) assume that $f(x)$ is a sparse collection of $k$-degree monomials on the Boolean hypercube and apply sparse recovery to achieve a $O(n^k)$ runtime bound. We will show that under the case that $f(x) = g(\mathbf{P_A}x)$, our algorithm will inherently pick up any low-dimensional structure in $f(x)$ and achieve a convergence rate that depends on $k \log(n)$. This initial convergence rate survives, even if we perturb $f(x) = g(\mathbf{P_A}x) + h(x)$, so long as $h(x)$ is sufficiently small.

We will not cover the whole variety of black-box optimization methods, such as Bayesian optimization or genetic algorithms. In general, these methods attempt to solve a broader problem (e.g. multiple optima), have weaker theoretical guarantees and may require substantial computation at each step: e.g. Bayesian optimization generally has theoretical iteration complexities that grow exponentially in dimension, and CMA-ES lacks provable complexity bounds beyond convex quadratic functions. In addition to the slow runtime and weaker guarantees, Bayesian optimization assumes the success of an inner optimization loop of the acquisition function. This inner optimization is often implemented with many iterations of a simpler zeroth-order methods, justifying the need to understand gradient-less descent algorithms within its own context.

## 1.1 Our contributions

In this paper, we present GradientLess Descent (GLD), a class of truly gradient-free algorithms (also known as direct search algorithms) that are parameter free and provably fast. Our algorithms are based on a simple intuition: for well-conditioned functions, if we start from a point and take a small step in a randomly chosen direction, there is a significant probability that we will reduce the objective function value. We present a novel analysis that relies on facts in high dimensional geometry and can thus be viewed as a geometric analysis of gradient-free algorithms, recovering the standard convergence rates and step sizes. Specifically, we show that if the step size is on the order of $O(\frac{1}{\sqrt{n}})$, we can guarantee an expected decrease of $1 - \Omega(\frac{1}{n})$ in the optimality gap, based on geometric properties of the sublevel sets of a smooth and strongly convex function.

Our results are invariant under monotone transformations of the objective function, thus our convergence results also hold for a large class of non-convex functions that are a subclass of quasi-convex functions. Specifically, note that monotone transformations of convex functions are not necessarily convex. However, a monotone transformation of a convex function is always *quasi-convex*. The maximization of *quasi-concave* utility functions, which is equivalent to the minimization of quasi-convex functions, is an important topic of study in economics (e.g. Arrow & Enthoven (1961)).

Table 1: Comparison of zeroth order optimization for well-conditioned convex functions where $R = \|x_0 - x^*\|$ and $F = f(x_0) - f(x^*)$. 'Monotone' column indicates the invariance under monotone transformations (Definition 4). 'k-Sparse' and 'k-Affine' columns indicate that iteration complexity is poly($k$, $\log(n)$) when $f(x)$ depends only on a k-sparse subset of coordinates or on a rank-$k$ affine subspace.

| Algorithm | Iteration Complexity | Monotone | k-Sparse | k-Affine |
|---|---|---|---|---|
| Nesterov & Spokoiny (2011) | $n\log((R^2 + F)/\epsilon)$ | No | No | No |
| Balasubramanian & Ghadimi (2018) | $n\log(F/\epsilon)$ | No | Yes | No |
| Stich et al. (2013) | $n\log(F/\epsilon)$ | Yes | No | No |
| Gorbunov et al. (2019) | $n\log(F/\epsilon)$ | Yes | No | No |
| This paper (GLD) | $n\log(R/\epsilon)$ | Yes | Yes | Yes |

Intuition suggests that the step-size dependence on dimensionality can be improved when $f(x)$ admits a low-dimensional structure. With a careful choice of sampling distribution we can show that if $f(x) = g(\mathbf{P_A}x)$, where $\mathbf{P_A}$ is a rank $k$ matrix, then our step size can be on the order of $O(\frac{1}{\sqrt{k}})$ as our optimization behavior is preserved under projections. We call this property affine-invariance and show that the number of function evaluations needed for convergence depends logarithmically on $n$. Unlike most previous algorithms in the high-dimensional setting, no expensive sparse recovery or subspace finding methods are needed. Furthermore, by novel perturbation arguments, we show that our fast convergence rates are *robust* and holds even under the more realistic assumption when $f(x) = g(\mathbf{P_A}x) + h(x)$ with $h(x)$ being sufficiently small.

**Theorem 1** (Convergence of GLD: Informal Restatement of Theorem 7 and Theorem 14). *Let $f(x)$ be any monotone transform of a convex function with condition number $Q$ and $R = \|x_0 - x^*\|$. Let $y$ be a sample from an appropriate distribution centered at $x$. Then, with constant probability,*

$$f(y) - f(x^*) \leq (f(x) - f(x^*)) \left(1 - \tfrac{1}{5nQ}\right)$$

*Therefore, we can find $x_T$ such that $\|x_T - x^*\| \leq \epsilon$ after $T = \widetilde{O}(nQ\log(R/\epsilon))$ function evaluations. Furthermore, for functions $f(x) = g(\mathbf{P_A}x) + h(x)$ with rank $k$ matrix $\mathbf{P_A}$ and sufficiently small $h(x)$, we only require $\widetilde{O}(kQ\log(n)\log(R/\epsilon))$ evaluations.*

Another advantage of our non-standard geometric analysis is that it allows us to deduce that our rates are optimal with a matching lower bound (up to logarithmic factors), presenting theoretical evidence that gradient-free inherently requires $\Omega(nQ)$ function evaluations to converge. While gradient-estimation algorithms can achieve a better theoretical iteration complexity of $O(n\sqrt{Q})$, they lack the monotone and affine invariance properties. Empirically, we see that invariance properties are important to successful optimization, as validated by experiments on synthetic BBOB and MuJoCo benchmarks that show the competitiveness of GLD against standard optimization procedures.

## 2 PRELIMINARIES

We first define a few notations for the rest of the paper. Let $\mathcal{X}$ be a compact subset of $\mathbb{R}^n$ and let $\|\cdot\|$ denote the Euclidean norm. The diameter of $\mathcal{X}$, denoted $\|\mathcal{X}\| = \max_{x,x' \in \mathcal{X}} \|x - x'\|$, is the maximum distance between elements in $\mathcal{X}$. Let $f : \mathcal{X} \to \mathbb{R}$ be a real-valued function which attains its minimum at $x^*$. We use $f(\mathcal{X}) = \{f(x) : x \in \mathcal{X}\}$ to denote the image of $f$ on a subset $\mathcal{X}$ of $\mathbb{R}^n$, and $\mathcal{B}(c, r) = \{x \in \mathbb{R}^n : \|c - x\| \leq r\}$ to denote the ball of radius $r$ centered at $c$.

**Definition 2.** *The level set of $f$ at point $x \in \mathcal{X}$ is $\mathcal{L}_c(f) = \{y \in \mathcal{X} : f(y) = f(x)\}$. The sub-level set of $f$ at point $x \in \mathcal{X}$ is $\mathcal{L}_c^{\downarrow}(f) = \{y \in \mathcal{X} : f(y) \leq f(x)\}$. When the function $f$ is clear from the context, we omit it.*

**Definition 3.** *We say that $f$ is $\alpha$-strongly convex for $\alpha > 0$ if $f(y) \geq f(x) + \langle\nabla f(x), y - x\rangle + \frac{\alpha}{2}\|y - x\|^2$ for all $x, y \in \mathcal{X}$ and $\beta$-smooth for $\beta > 0$ if $f(y) \leq f(x) + \langle\nabla f(x), y - x\rangle + \frac{\beta}{2}\|y - x\|^2$ for all $x, y \in \mathcal{X}$.*

**Definition 4.** *We say that $g \circ f$ is a* monotone transformation *of $f$ if $g : f(\mathcal{X}) \to \mathbb{R}$ is a monotonically (and strictly) increasing function.*

Monotone transformations preserve the level sets of a function in the sense that $\mathcal{L}_x(f) = \mathcal{L}_x(g \circ f)$. Because our algorithms depend only on the level set properties, our results generalize to any monotone transformation of a strongly convex and strongly smooth function. This leads to our extended notion of condition number.

**Definition 5.** *A function $f$ has condition number $Q \geq 1$ if it is the minimum ratio $\beta/\alpha$ over all functions $g$ such that $f$ is a monotone transformation of $g$ and $g$ is $\alpha$-strongly convex and $\beta$ smooth.*

When we work with low rank extensions of $f$, we only care about the condition number of $f$ within a rank $k$ subspace. Indeed, if $f$ only varies along a rank $k$ subspace, then it has a strong convexity value of 0, making its condition number undefined. If $f$ is $\alpha$-strongly convex and $\beta$-smooth, then its Hessian matrix always has eigenvalues bounded between $\alpha$ and $\beta$. Therefore, we need a notion of a projected condition number. Let $\mathbf{A} \in \mathbb{R}^{d \times k}$ be some orthonormal matrix and let $\mathbf{P_A} = \mathbf{AA}^\top$ be the projection matrix onto the column space of $\mathbf{A}$.

**Definition 6.** *For some orthonormal $\mathbf{A} \in \mathbb{R}^{d \times k}$ with $d > k$, a function $f$ has condition number restricted to $\mathbf{A}$, $Q(\mathbf{A}) \geq 1$, if it is the minimum ratio $\beta/\alpha$ over all functions $g$ such that $f$ is a monotone transformation of $g$ and $h(y) = g(\mathbf{A}y)$ is $\alpha$-strongly convex and $\beta$ smooth.*

## 3 ANALYSIS OF DESCENT STEPS

The GLD template can be summarized as follows: given a sampling distribution $\mathcal{D}$, we start at $x_0$ and in iteration $t$, we choose a scalar radii $r_t$ and we sample $y_t$ from a distribution $r_t \mathcal{D}$ centered around $x_t$, where $r_t$ provides the scaling of $\mathcal{D}$. Then, if $f(y_t) < f(x_t)$, we update $x_{t+1} = y_t$; otherwise, we set $x_{t+1} = x_t$. The analysis of GLD follows from the main observation that the sub-level set of a monotone transformation of a strongly convex and strongly smooth function contains a ball of sufficiently large radius tangent to the level set (Lemma 15). In this section, we show that this property, combined with facts of high-dimensional geometry, implies that moving in a random direction from any point has a good chance of significantly improving the objective.

As we mentioned before, the key to fast convergence is the careful choice of step sizes, which we describe in Theorem 7. The intuition here is that we would like to take as large steps as possible while keeping the probability of improving the objective function reasonably high, so by insights in high-dimensional geometry, we choose a step size of $\Theta(1/\sqrt{n})$. Also, we show that if $f(x)$ admits a latent rank-$k$ structure, then this step size can be increased to $\Theta(1/\sqrt{k})$ and is therefore only dependent on the latent dimensionality of $f(x)$, allowing for fast high-dimensional optimization. Lastly, our geometric understanding allows us to show that our convergence rates are optimal with a matching lower bound. Without loss of generality, this section assumes that $f(x)$ is strongly convex and smooth with condition number $Q$.

### 3.1 STEP SIZE

**Theorem 7.** *For any $x$ such that $\frac{3}{5Q}\|x - x^*\| \in [C_1, C_2]$, we can find integers $0 \leq k_1, k_2 < \log \frac{C_2}{C_1}$ such that if $r = 2^{k_1}C_1$ or $r = 2^{-k_2}C_2$, then a random sample $y$ from uniform distribution over $B_x = \mathcal{B}(x, \frac{r}{\sqrt{n}})$ satisfies*

$$f(y) - f(x^*) \leq (f(x) - f(x^*))\left(1 - \frac{1}{5nQ}\right)$$

*with probability at least $\frac{1}{4}$.*

Proving the above theorem requires the following lemma about the intersection of balls in high dimensions and it is proved in the appendix.

**Lemma 8.** *Let $B_1$ and $B_2$ be two balls in $\mathbb{R}^n$ of radii $r_1$ and $r_2$ respectively. Let $\ell$ be the distance between the centers. If $r_1 \in [\frac{\ell}{2\sqrt{n}}, \frac{\ell}{\sqrt{n}}]$ and $r_2 \geq \ell - \frac{\ell}{4n}$, then*

$$\mathrm{vol}\,(B_1 \cap B_2) \geq c_n \,\mathrm{vol}\,(B_1),$$

*where $c_n$ is a dimension-dependent constant that is lower bounded by $\frac{1}{4}$ at $n = 1$.*

## 3.2 Gaussian Sampling and Low Rank Structure

A direct application of Lemma 8 seems to imply that uniform sampling of a high-dimensional ball is necessary. Upon further inspection, this can be easily replaced with a much simpler Gaussian sampling procedure that concentrates the mass close to the surface to the ball. This procedure lends itself to better analysis when $f(x)$ admits a latent low-dimensional structure since any affine projection of a Gaussian is still Gaussian.

**Lemma 9.** *Let $B_1$ and $B_2$ be two balls in $\mathbb{R}^n$ of radii $r_1$ and $r_2$ respectively. Let $\ell$ be the distance between the centers. If $r_1 \in [\frac{\ell}{2\sqrt{n}}, \frac{\ell}{\sqrt{n}}]$ and $r_2 \geq \ell - \frac{\ell}{n}$ and $X = (X_1, ..., X_n)$ are independent Gaussians with mean centered at the center of $B_1$ and variance $\frac{r_1^2}{n}$, then*

$$\mathbf{Pr}[X \in B_2] > c,$$

*where $c$ is a dimension-independent constant.*

Assume that there exists some rank $k$ projection matrix $\mathbf{P_A}$ such that $f(x) = g(\mathbf{P_A}x)$, where $k$ is much smaller than $n$. Because Gaussians projected on a $k$-dimensional subspace are still Gaussians, we show that our algorithm has a dimension dependence on $k$. We let $Q_g(\mathbf{A})$ be the condition number of $g$ restricted to the subspace $\mathbf{A}$ that drives the dominant changes in $f(x)$.

**Theorem 10.** *Let $f(x) = g(\mathbf{P_A}x)$ for some unknown rank $k$ matrix $\mathbf{P_A}$ with $k < n$ and suppose $\frac{3}{5Q}\|\mathbf{P_A}(x - x^*)\| \in [C_1, C_2]$ for some numbers $C_1, C_2 \in \mathbb{R}^+$. Then, there exist integers $0 \leq k_1, k_2 < \log \frac{C_2}{C_1}$ such that if $r = 2^{k_1}C_1$ or $r = 2^{-k_2}C_2$, then a random sample $y$ from a Gaussian distribution $\mathcal{N}(x, \frac{r^2}{k}\mathbf{I})$ satisfies*

$$f(y) - f(x^*) \leq (f(x) - f(x^*)) \left(1 - \frac{1}{5kQ_g(\mathbf{A})}\right)$$

*with constant probability.*

Note that the speed-up in progress is due to the fact that we can now tolerate the larger sampling radius of $\Omega(1/\sqrt{k})$, while maintaining a high probability of making progress. If $k$ is unknown, we can simply use binary search to find the correct radius with an extra factor of $\log(n)$ in our runtime.

The low-rank assumption is too restrictive to be realistic; however, our fast rates still hold, at least for the early stages of the optimization, even if we assume that $f(x) = g(\mathbf{P_A}x) + h(x)$ and $|h(x)| \leq \delta$ is a full-rank function that is bounded by $\delta$. In this setting, we can show that convergence remains fast, at least until the optimality gap approaches $\delta$.

**Theorem 11.** *Let $f(x) = g(\mathbf{P_A}x) + h(x)$ for some unknown rank $k$ matrix $\mathbf{P_A}$ with $k < n$ where $g, h$ are convex and $|h| \leq \delta$. Suppose $\frac{3}{5Q}\|\mathbf{P_A}x - z^*\| \in [C_1, C_2]$ for some numbers $C_1, C_2 \in \mathbb{R}^+$ where $z^*$ minimizes $g(z)$. Then, there exist integers $0 \leq k_1, k_2 < \log \frac{C_2}{C_1}$ such that if $r = 2^{k_1}C_1$ or $r = 2^{-k_2}C_2$, then a random sample $y$ from a Gaussian distribution $\mathcal{N}(x, \frac{r^2}{k}\mathbf{I})$ satisfies*

$$f(y) - f(x^*) \leq (f(x) - f(x^*)) \left(1 - \frac{1}{10kQ_g(\mathbf{A})}\right)$$

*with constant probability whenever $f(x) - f(x^*) \geq 60\delta k Q_g(\mathbf{A})$.*

## 3.3 Lower Bounds

We show that our upper bounds given in the previous section are tight up to logarithmic factors for any symmetric sampling distribution $\mathcal{D}$. These lower bounds are easily derived from our geometric perspective as we show that a sampling distribution with a large radius gives an extremely low probability of intersection with the desired sub-level set. Therefore, while gradient-approximation algorithms can be accelerated to achieve a runtime that depends on the square-root of the condition number $Q$, gradient-less methods that rely on random sampling are likely unable to be accelerated according to our lower bound. However, we emphasize that monotone invariance allows these results to apply to a broader class of objective functions, beyond smooth and convex, so the results can be useful in practice despite the seemingly worse theoretical bounds.

---

**Algorithm 1:** Gradientless Descent with Binary Search (GLD-Search)

---

**Input:** function: $f : \mathbb{R}^n \to \mathbb{R}$, $T \in \mathbb{Z}_+$: number of iterations, $x_0$: starting point,
$\quad\quad$ $\mathcal{D}$: sampling distribution, $R$: maximum search radius, $r$: minimum search radius

1 Set $K = \log(R/r)$
2 **for** $t = 0, \ldots, T$ **do**
3 $\quad$ **Ball Sampling Trial** $i$**:**
4 $\quad$ **for** $k = 0, \ldots, K$ **do**
5 $\quad\quad$ Set $r_{i,k} = 2^{-k}R$.
6 $\quad\quad$ Sample $v_{i,k} \sim r_{i,k}\mathcal{D}$.
7 $\quad$ **end**
8 $\quad$ **Update:** $x_{t+1} = \arg\min_k \left\{ f(y) \,\Big|\, y = x_t,\ y = x_t + v_{i,k} \right\}$
9 **end**
10 **return** $x_t$

---

**Theorem 12.** *Let $y = x + v$, where $v$ is a random sample from $r\mathcal{D}$ for some radius $r > 0$ and $\mathcal{D}$ is standard Gaussian or any rotationally symmetric distribution. Then, there exist a region $X$ with positive measure such that for any $x \in X$,*

$$f(y) - f(x^*) \geq (f(x) - f(x^*))\left(1 - \frac{\sqrt{5\log(nQ)}}{nQ}\right)$$

*with probability at least $1 - \frac{1}{poly(nQ)}$.*

## 4 GRADIENTLESS ALGORITHMS

In this section, we present two algorithms that follow the same Gradientless Descent (GLD) template: GLD-Search and GLD-Fast, with the latter being an optimized version of the former when an upper bound on the condition number of a function is known. For both algorithms, since they are monotone-invariant, we appeal to the previous section to derive fast convergence rates for any monotone transform of convex $f(x)$ with good condition number. We show the efficacy of both algorithms experimentally in the Experiments section.

### 4.1 GRADIENTLESS DESCENT WITH BINARY SEARCH

Although the sampling distribution $\mathcal{D}$ is fixed, we have a choice of radii for each iteration of the algorithm. We can apply a binary search procedure to ensure progress. The most straightforward version of our algorithm is thus with a naive binary sweep across an interval in $[r, R]$ that is unchanged throughout the algorithm. This allows us to give convergence guarantees without previous knowledge of the condition number at a cost of an extra factor of $\log(n/\epsilon)$.

**Theorem 13.** *Let $x_0$ be any starting point and $f$ a blackbox function with condition number $Q$. Running Algorithm 1 with $r = \frac{\epsilon}{\sqrt{n}}$, $R = \|\mathcal{X}\|$ and $\mathcal{D} = \mathcal{N}(0, \mathbf{I})$ as a standard Gaussian returns a point $x_T$ such that $\|x_T - x^*\| \leq 2Q^{3/2}\epsilon$ after $O(nQ\log(n\|\mathcal{X}\|/\epsilon)^2)$ function evaluations with high probability.*

*Furthermore, if $f(x) = g(\mathbf{P_A}x)$ admits a low-rank structure with $\mathbf{P_A}$ a rank $k$ matrix, then we only require $O(kQ_g(\mathbf{A})\log(n\|\mathcal{X}\|/\epsilon)^2)$ function evaluations to guarantee $\|\mathbf{P_A}(x_T - x^*)\| \leq \epsilon$. This holds analogously even if $f(x) = g(\mathbf{P_A}x) + h(x)$ is almost low-rank where $|h| \leq \delta$ and $\epsilon > 60\delta k Q_g(\mathbf{A})$.*

### 4.2 GRADIENTLESS DESCENT WITH FAST BINARY SEARCH

GLD-Search (Algorithm 1) uses a naive lower and upper bound for the search radius $\|x_t - x^*\|$, which incurs an extra factor of $\log(1/\epsilon)$ in the runtime bound. In GLD-Fast, we remove this extra factor dependence on $\log(1/\epsilon)$ by drastically reducing the range of the binary search. This is done by exploiting the assumption that $f$ has a good condition number upper bound $\hat{Q}$ and by slowly halfing the diameter of the search space every few iterations since we expect $x_t \to x^*$ as $t \to \infty$.

---

**Algorithm 2:** Gradientless Descent with Fast Binary Search (GLD-Fast)

---

**Input:** function $f : \mathbb{R}^n \to \mathbb{R}$, $T \in \mathbb{Z}_+$: number of iterations, $x_0$: starting point,
$\qquad$ $\mathcal{D}$: sampling distribution, $R$: diameter of search space, $Q$: condition number bound

1 Set $K = \log(4\sqrt{Q})$, $H = nQ\log(Q)$
2 **for** $t = 1, \dots, T$ **do**
3 $\quad$ Set $R = R/2$ when $T \equiv 0 \mod H$ (every $H$ iterations).
4 $\quad$ **Ball Sampling Trial** $i$:
5 $\quad$ **for** $k$ = -$K$, $\dots$, $0$, $\dots$, $K$ **do**
6 $\quad\quad$ Set $r_{i,k} = 2^{-k}R$.
7 $\quad\quad$ Sample $v_{i,k} \sim r_{i,k}\mathcal{D}$.
8 $\quad$ **end**
9 $\quad$ **Update:** $x_{t+1} = \arg\min_i \left\{ f(y) \middle| y = x_t, \, y = x_t + v_i \right\}$
10 **end**
11 **return** $x_t$

---

**Theorem 14.** *Let $x_0$ be any starting point and $f$ a blackbox function with condition number upper bounded by $Q$. Running Algorithm 2 with suitable parameters returns a point $x_T$ such that $f(x_T) - f(x^*) \leq \epsilon$ after $O(nQ\log^2(Q)\log(\|\mathcal{X}\|/\epsilon))$ function evaluations with high probability.*

*Furthermore, if $f(x) = g(\mathbf{P_A}x)$ admits a low-rank structure with $\mathbf{P_A}$ a rank $k$ matrix, then we only require $O(kQ_g(\mathbf{A})\log(n)\log^2(Q_g(\mathbf{A}))\log(\|\mathcal{X}\|/\epsilon))$ function evaluations to guarantee $\|\mathbf{P_A}(x_T - x^*)\| \leq \epsilon$. This holds analogously even if $f(x) = g(\mathbf{P_A}x) + h(x)$ is almost low-rank where $|h| \leq \delta$ and $\epsilon > 60\delta k Q_g(\mathbf{A})$.*

## 5 EXPERIMENTS

We tested GLD algorithms on a simple class of objective functions and compare it to Accelerated Random Search (ARS) by Nesterov & Spokoiny (2011), which has linear convergence guarantees on strongly convex and strongly smooth functions. To our knowledge, ARS makes the weakest assumption among the zeroth-order algorithms that have linear convergence guarantees and perform only a constant order of operations per iteration. Our main conclusion is that GLD-Fast is comparable to ARS and tends to achieve a reasonably low error much faster than ARS in high dimensions ($\geq 50$). In low dimensions, GLD-Search is competitive with GLD-Fast and ARS though it requires no information about the function.

We let $H_{\alpha,\beta,n} \in \mathbb{R}^{n \times n}$ be a diagonal matrix with its $i$-th diagonal equal to $\alpha + (\beta - \alpha)\frac{i-1}{n-1}$. In simple words, its diagonal elements form an evenly space sequence of numbers from $\alpha$ to $\beta$. Our objective function is then $f_{\alpha,\beta,n} : \mathbb{R}^n \to \mathbb{R}$ as $f_{\alpha,\beta,n}(x) = \frac{1}{2}x^\top H_{\alpha,\beta,n}x$, which is $\alpha$-strongly convex and $\beta$-strongly smooth. We always use the same starting point $x = \frac{1}{\sqrt{n}}(1, \dots, 1)$, which requires $\|X\| = \sqrt{Q}$ for our algorithms. We plot the optimality gap $f(b_t) - f(x^*)$ against the number of function evaluations, where $b_t$ is the best point observed so far after $t$ evaluations. Although all tested algorithms are stochastic, they have a low variance on the objective functions that we use; hence we average the results over 10 runs and omit the error bars in the plots.

We ran experiments on $f_{1,8,n}$ with imperfect curvature information $\hat{\alpha}$ and $\hat{\beta}$ (see Figure 3 in appendix). GLD-Search is independent of the condition number. GLD-Fast takes only one parameter, which is the upper bound on the condition number; if approximation factor is $z$, then we pass $8z$ as the upper bound. ARS requires both strong convexity and smoothness parameters. We test three different distributions of the approximation error; when the approximation factor is $z$, then ARS-alpha gets $(\alpha/z, \beta)$, ARS-beta gets $(\alpha, z\beta)$, and ARS-even gets $(\alpha/\sqrt{z}, \sqrt{z}\beta)$ as input. GLD-Fast is more robust and faster than ARS when the condition number is over-approximated. When the condition number is underestimated, GLD-Fast still steadily converges.

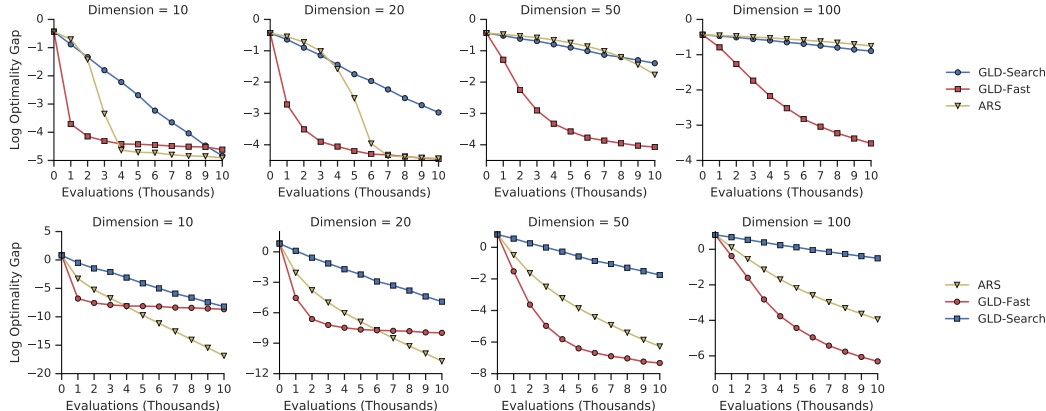

Figure 1: The average optimality gap on a quadratic objective function that is strongly convex and smooth objective (top); and its monotone transformation (bottom). Further experiments on non-convex BBOB functions show similar behavior and are in the appendix.

## 5.1 Monotone Transformations

In Figure 1, we ran experiments on $f_{1,8,n}$ for different settings of dimensionality $n$, and its monotone transformation with $g(y) = -\exp(-\sqrt{y})$. For this experiment, we assume a perfect oracle for the strong convexity and smoothness parameters of $f$. The convergence of GLD is totally unaffected by the monotone transformation. For the low-dimension cases of a transformed function (bottom half of the figure), we note that there are inflection points in the convergence curve of ARS. This means that ARS initially struggles to gain momentum and then struggles to stop the momentum when it gets close to the optimum. Another observation is that unlike ARS that needs to build up momentum, GLD-Fast starts from a large radius and therefore achieves a reasonably low error much faster than ARS, especially in higher dimensions.

## 5.2 BBOB Benchmarks

To show that practicality of GLD on practical and non-convex settings, we also test GLD algorithms on a variety of BlackBox Optimization Benchmarking (BBOB) functions (Hansen et al., 2009). For each function, the optima is known and we use the log optimality gap as a measure of competance. Because each function can exhibit varying forms of non-smoothness and convexity, all algorithms are ran with a smoothness constant of 10 and a strong convexity constant of 0.1. All other setup details are same as before, such as using a fixed starting point.

The plots, given in Appendix C, underscore the superior performance of GLD algorithms on various BBOB functions, demonstrating that GLD can successfully optimize a diverse set of functions even without explicit knowledge of condition number. We note that BBOB functions are far from convex and smooth, many exhibiting high conditioning, multi-modal valleys, and weak global structure. Due to our radius search produce, our algorithm appears more robust to non-ideal settings with non-convexity and ill conditioning. As expected, we note that GLD-Fast tend to outperform GLD-Search, especially as the dimension increases, matching our theoretical understanding of GLD.

## 5.3 Mujoco Control Benchmarks and Affine Transformations

We also ran experiments on the Mujoco benchmarks with varying architectures, both linear and nonlinear. This demonstrates the viability of our approach even in the non-convex, high dimensional setting. We note that however, unlike e.g. ES which uses all queries to form a gradient direction, our algorithm removes queries which produce less reward than using the current arg-max, which can be an information handicap. Nevertheless, we see that our algorithm still achieves competitive performance on the maximum reward. We used a horizon of 1000 for all experiments.

We further tested the affine invariance of GLD on the policy parameters from using Gaussian ball sampling, under the HalfCheetah benchmark by projecting the state $s$ of the MDP with linear policy to a higher dimensional state $Ws$, using a matrix multiplication with an orthonormal $W$. Specifically, in this setting, for a linear policy parametrized by matrix $K$, the objective function is thus $J(KW)$ where $\pi_K(Ws) = KWs$. Note that when projecting into a high dimension, there is a slowdown factor of $\log \frac{d_{new}}{d_{old}}$ where $d_{new}, d_{old}$ are the new high dimension and previous base dimension, respectively, due to the binary search in our algorithm on a higher dimensional space. For our HalfCheetah case, we projected the 17 base dimension to a 200-length dimension, which suggests that the slowdown factor is a factor $\log \frac{200}{17} \approx 3.5$. This can be shown in our plots in the appendix (Figure 15).

Table 2: Final rewards by GLD with linear (L) and deep (H41) policies on Mujoco Benchmarks show that GLD is competitive. We apply an affine projection on HalfCheetah to test affine invariance. We use the reward threshold found from (Mania et al., 2018) with Reacher's threshold (Schulman et al., 2017) for a reasonable baseline.

| Env. | Arch | Rew. at $(10^4, 10^5, \text{Max})$ Queries | Rew. Thresh. |
|---|---|---|---|
| HalfCheetah-v1 | L | 3799, 3903, 4064 | 3430 |
| HalfCheetah-v1, Proj-200 | L | 1594 , 3509 , 4342 | 3430 |
| HalfCheetah-v1 | H41 | 2741, 3074, 3392 | 3430 |
| Hopper-v1 | L | 1017, 3359, 3375 | 3120 |
| Hopper-v1 | H41 | 2708, 3370, 3566 | 3120 |
| Reacher-v1 | L | -70, -5, -4 | -10 |
| Reacher-v1 | H41 | -231, -17, -15 | -10 |
| Swimmer-v1 | L | 365, 369 , 369 | 325 |
| Swimmer-v1 | H41 | 353, 369, 369 | 325 |
| Walker2d-v1 | L | 1027 , 2201, 2201 | 4390 |
| Walker2d-v1 | H41 | 1630, 1963, 2146 | 4390 |

## 6 CONCLUSION

We introduced GLD, a robust zeroth-order optimization algorithm that is simple, efficient, and we show strong theoretical convergence bounds via our novel geometric analysis. As demonstrated by our experiments on BBOB and MuJoCo benchmarks, GLD performs very robustly even in the non-convex setting and its monotone and affine invariance properties give theoretical insight on its practical efficiency.

GLD is very flexible and allows easy modifications. For example, it could use momentum terms to keep moving in the same direction that improved the objective, or sample from adaptively chosen ellipsoids similarly to adaptive gradient methods. (Duchi et al., 2011; McMahan & Streeter, 2010). Just as one may decay or adaptively vary learning rates for gradient descent, one might use a similar change the distribution from which the ball-sampling radii are chosen, perhaps shrinking the minimum radius as the algorithm progresses, or concentrating more probability mass on smaller radii.

Likewise, GLD could be combined with random restarts or other restart policies developed for gradient descent. Analogously to adaptive per–coordinate learning rates Duchi et al. (2011); McMahan & Streeter (2010), one could adaptively change the shape of the balls being sampled into ellipsoids with various length-scale factors. Arbitrary combinations of the above variants are also possible.

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

## A  PROOFS OF SECTION 3

**Lemma 15.** *If $h$ has condition number $Q$, then for all $x \in \mathcal{X}$, there is a ball of radius $Q^{-1}\|x - x^*\|$ that is tangent at $x$ and inside the sublevel set $\mathcal{L}_x^{\downarrow}(h)$.*

*Proof.* Write $h = g \circ f$ such that $f$ is $\alpha$-strongly convex and $\beta$-smooth for some $\beta = Q\alpha$ and $g$ is monotonically increasing. From the smoothness assumption, we have for any $s$,

$$
f\left(x - \tfrac{1}{\beta}\nabla f(x) + s\right)
$$
$$
\leq f(x) + \left\langle \nabla f(x), s - \tfrac{1}{\beta}\nabla f(x)\right\rangle + \tfrac{\beta}{2}\left\|s - \tfrac{1}{\beta}\nabla f(x)\right\|^2
$$
$$
= f(x) + \tfrac{\beta}{2}\left(\|s\|^2 - \tfrac{1}{\beta^2}\|\nabla f(x)\|^2\right).
$$

Consider the ball $B = \mathcal{B}(x - \tfrac{1}{\beta}\nabla f(x), \tfrac{1}{\beta}\|\nabla f(x)\|)$. For any $y \in B$, the above inequality implies $f(y) \leq f(x)$. Hence, when we apply $g$ on both sides, we still have $h(y) \leq h(x)$ for all $y \in B$. Therefore, $B \subseteq \mathcal{L}_{h(y)}^{\downarrow}$.

By strong convexity, $\|\nabla f(x)\| \geq \alpha\|x - x^*\|$. It follows that the radius of $B$ is at least $\tfrac{\alpha}{\beta}\|x - x^*\|$.  $\square$

*Proof of Lemma 8.* Without loss of generality, consider the unit distance case where $\ell = 1$. Furthermore, it suffices to prove for the smallest possible radius $r_2 = 1 - \tfrac{1}{4n}$.

Since $|r_1 - r_2| \leq \ell \leq r_1 + r_2$, the intersection $B_1 \cap B_2$ is composed of two hyperspherical caps glued end to end. We lower bound $\text{vol}(B_1 \cap B_2)$ by the volume of the cap $C_1$ of $B_1$ that is contained in the intersection. Consider the triangle with sides $r_1, r_2$ and $\ell$. From classic geometry, the height of $C_1$ is

$$
c_1 = \tfrac{1}{2}\left(1 + r_1^2 - r_2^2\right) > 0. \tag{1}
$$

The volume of a spherical cap is Li (2011),

$$
\text{vol}(C_1) = \frac{1}{2}\,\text{vol}(B_1)\,I_{1 - \frac{c_1^2}{r_1^2}}\left(\frac{n+1}{2}, \frac{1}{2}\right).
$$

where $I$ is the regularized incomplete beta function defined as

$$
I_x(a, b) = \frac{\int_0^x t^{a-1}(1-t)^{b-1}\, dt}{\int_0^1 t^{a-1}(1-t)^{b-1}\, dt}
$$

where $x \in [0, 1]$ and $a, b \in (0, \infty)$. Note that for any fixed $a$ and $b$, $I_x(a, b)$ is increasing in $x$. Hence, in order to obtain a lower bound on $\text{vol}(C_1)$, we want to lower bound $1 - \frac{c_1^2}{r_1^2}$ or equivalently, upper bound $\frac{c_1^2}{r_1^2}$.

Write $r_1 = \frac{\alpha}{2\sqrt{n}}$ for some $\alpha \in [1, 2]$. From Eq. (1),

$$
c_1 = \frac{1}{4n} + \frac{\alpha^2}{8n} - \frac{1}{32n^2}.
$$

Hence,

$$
\frac{c_1}{r_1} = \frac{1}{16\sqrt{n}}\left(\frac{8}{\alpha} + 4\alpha - \frac{1}{n}\right)
$$

Since $g(\alpha) = \frac{8}{\alpha} + 4\alpha$ is convex in $[1, 2]$, $g(\alpha) \leq \max(g(1), g(2)) = 12$. It follows that $\frac{c_1}{r_1} \leq \frac{1}{16\sqrt{n}}\left(12 - \frac{1}{n}\right) \leq \frac{3}{4\sqrt{n}}$. So, $1 - \frac{c_1^2}{r_1^2} \geq 1 - \frac{9}{16n}$. To complete the proof, note that $V_n := I_{1 - \frac{9}{16n}}\left(\frac{n+1}{2}, \frac{1}{2}\right)$ is increasing in $n$, and $V_1 = \frac{1}{4}$. As $n$ goes to infinity, this value converges to 1 as $B_1 \subset B_2$.  $\square$

*Proof of Lemma 7.* Let $\nu = \frac{1}{5nQ}$. Let $q = (1-\nu)x + \nu x^*$. Let $B_q = \mathcal{B}(c_q, r_q)$ be a ball that has $q$ on its surface, lies inside $\mathcal{L}_q^{\downarrow}$, and has radius $r_q = Q^{-1}\|x - x^*\|$. Lemma 15 guarantees its existence.

Suppose that

$$\text{vol}\left(B_x \cap B_q\right) \geq \tfrac{1}{4}\,\text{vol}\left(B_x\right) \tag{2}$$

and that a random sample $y$ from $B_x$ belongs to $B_q$, which happens with probability at least $\frac{1}{4}$. Then, our guarantee follows by

$$
\begin{aligned}
f(y) - f(x^*) &\leq f(q) - f(x^*) \\
&\leq (1-\nu)f(x) + \nu f(x^*) - f(x^*) \\
&\leq (1-\nu)\left(f(x) - f(x^*)\right)
\end{aligned}
$$

where the first line follows from Lemma 15 and second line from convexity of $f$.

Therefore, it now suffices to prove Eq. 2. To do so, we will apply Lemma 8 after showing that the radius of $B_x$ and $B_q$ are in the proper ranges. Let $\ell = \|x - c_q\|$ and note that

$$
\begin{aligned}
\ell &\leq \|x - q\| + r_q \tag{3} \\
&\leq \nu\|x - x^*\| + r_q = \nu\|x - x^*\| + Q^{-1}\|q - x^*\| \\
&\leq \left(\nu + Q^{-1}(1 - \nu)\right)\|x - x^*\| \tag{4} \\
&\leq \tfrac{6}{5Q}\|b_x - x^*\|.
\end{aligned}
$$

Since $x$ is outside of $B_q$, we also have

$$
\begin{aligned}
\ell &\geq r_q = Q^{-1}\|q - x^*\| = Q^{-1}(1-\nu)\|x - x^*\| \\
&\geq \tfrac{4}{5Q}\|b_x - x^*\|. \tag{5}
\end{aligned}
$$

It follows that

$$\frac{\ell}{2} \leq \frac{3}{5Q}\|b_x - x^*\| \leq \ell.$$

In the $\log_2$ space, our choice of $k_1$ is equivalent to starting from $\log_2 C_1$ and sweeping through the range $[\log_2 C_1, \log_2 C_2]$ at the interval of size 1. This is guaranteed to find a point between $\frac{\ell}{2}$ and $\ell$, which is also an interval of size 1. Therefore, there exists a $k_1$ satisfying the theorem statement, and similarly, we can prove the existence of $k_2$.

Finally, it remains to show that $r_q \geq (1 - 1/(4n))\ell$. From Eq. (3), it suffices to show that $\|x - q\| \leq \frac{\ell}{4n}$ or equivalently $\nu\|x - x^*\| \leq \frac{\ell}{4n}$. From Eq. (4),

$$\|x - q\| = \nu\|x - x^*\| \leq \nu Q(1 - \nu)^{-1}\ell.$$

For any $Q, n \geq 1$, $1 - \nu \geq \frac{4}{5}$. So,

$$\nu Q(1 - \nu)^{-1} = \tfrac{1}{5n}(1 - \nu)^{-1} \leq \tfrac{1}{4n} \tag{6}$$

and the proof is complete. □

*Proof of Lemma 9.* Without loss of generality, let $\ell = 1$ and $B_2$ is centered at the origin with radius $r_2$ and $B_1$ is centered at $e_1 = (1, 0, ..., 0)$. Then, we simply want to show that

$$\mathbf{Pr}\left[(1 + X_1)^2 + \sum_{i=2}^{n} X_i^2 \leq r_2^2\right] > c_n$$

By Markov's inequality, we see that $\sum_{i=2}^{n} X_i^2 \leq 2r_1^2 = 2/n$ with probability at most $1/2$. And since $X_1$ is independent and $r_2 \geq 1 - 1/n$, it suffices to show that

$$\mathbf{Pr}\left[(1 + X_1)^2 \leq 1 - 4/n\right] > \Omega(1)$$

Since $X_1$ has standard deviation at least $r_1/\sqrt{n} \geq 1/(2n)$, we see that the probability of deviating at least a few standard deviation below is at least a constant. □

*Proof of Theorem 10.* We can consider the projection of all points onto the column space of $A$ and since the Gaussian sampling process is preserved, our proof follows from applying Theorem 7 restricted onto the $k$-dimensional subspace and using Lemma 9 in place of Lemma 8. $\qquad\square$

*Proof of Theorem 11.* By the boundedness of $h$, since $f(x) - f(x^*) \geq 60\delta k Q_g(\mathbf{A})$, we see that $g(\mathbf{P_A}x) - g(x^*) \geq 60\delta k Q_g(\mathbf{A}) - 2\delta > 0$. By Lemma 9, we see that if we sample from a Gaussian distribution $y \sim \mathcal{N}(x, \frac{r^2}{k}\mathbf{I})$, then if $z^*$ is the minimum of $g(x)$ restricted to the column space of $\mathbf{A}$, then

$$g(\mathbf{P_A}y) - g(z^*) \leq (g(\mathbf{P_A}x) - g(z^*))\left(1 - \frac{1}{5kQ_g(\mathbf{A})}\right)$$

with constant probability. By boundedness on $h$, we know that $h(y) \leq h(x) + 2\delta$. Furthermore, this also implies that $g(\mathbf{P_A}x^*) \leq g(z^*) + 2\delta$. Therefore, we know that the decrease is at least

$$
\begin{aligned}
f(y) - f(x^*) &= g(\mathbf{P_A}y) - g(\mathbf{P_A}x^*) + h(y) - h(x^*) \\
&\leq g(\mathbf{P_A}y) - g(z^*) + 2\delta \\
&\leq (g(\mathbf{P_A}x) - g(z^*))\left(1 - \frac{1}{5kQ_g(\mathbf{A})}\right) + 2\delta \\
&\leq (g(\mathbf{P_A}x) - g(\mathbf{P_A}x^*) + 2\delta)\left(1 - \frac{1}{5kQ_g(\mathbf{A})}\right) + 2\delta \\
&\leq (f(x) - f(x^*) + 4\delta)\left(1 - \frac{1}{5kQ_g(\mathbf{A})}\right) + 2\delta \\
&\leq (f(x) - f(x^*))\left(1 - \frac{1}{5kQ_g(\mathbf{A})}\right) + 6\delta
\end{aligned}
$$

Since $f(x) - f(x^*) \geq 10\delta k Q_g(A)$, we conclude that $(f(x) - f(x^*))\left(1 - \frac{1}{5kQ_g(\mathbf{A})}\right) + 6\delta \leq (f(x) - f(x^*))\left(1 - \frac{1}{10kQ_g(\mathbf{A})}\right)$ and our proof is complete. $\qquad\square$

*Proof of Theorem 12.* Our main proof strategy is to show that progress can only be made with a radius size of $O(\sqrt{\log(nQ)}/(nQ))$; larger radii cannot find descent directions with high probability. Consider a simple ellipsoid function $f(x) = x^\top D x$, where $D$ is a diagonal matrix and $D_{11} \leq D_{22} \leq ... \leq D_{nn}$, where WLOG we let $D_{11} = 1$ and $D_{ii} = Q$ for $i > 1$. The optima is $x^* = 0$ with $f(x^*) = 0$.

Consider the region $X = \{x = (x_1, x_2, ..., x_n) | 1 \geq x_1 \geq 0.9, |x_i| \leq 0.1/(Q\sqrt{n})\}$. Then, if we let $v \sim N(0, I)$ be a standard Gaussian vector, then for some radius $r$, we see that the probability of finding a descent direction is:

$$
\begin{aligned}
\mathbf{Pr}[f(x + rv) \leq f(x)] &= \mathbf{Pr}\left[(x_1 + rv_1)^2 + \sum_{i>1} D_{ii}(x_i + rv_i)^2 \leq x_1^2 + \sum_{i>1} D_{ii}x_i^2\right] \\
&= \mathbf{Pr}\left[2rx_1v_1 + r^2v_1^2 + Q\sum_{i>1}(2rx_iv_i + r^2v_i^2) \leq 0\right] \\
&\leq \mathbf{Pr}\left[2rx_1v_1 \leq -Q\sum_{i>1} 2rx_iv_i - Qr^2\sum_{i>1} v_i^2\right] \\
&= \mathbf{Pr}\left[v_1X_1 \leq -Q\sum_{i>1} x_iv_i - \frac{1}{2}Qr\sum_{i>1} v_i^2\right]
\end{aligned}
$$

By standard concentration bounds for sub-exponential variables, we have

$$\mathbf{Pr}\left[|\frac{1}{n-1}\sum_{i>1}v_i^2 - 1| \geq t\right] \leq 2e^{-(n-1)t^2/8}$$

Therefore, with exponentially high probability, $\sum_{i>1}X_i^2 \geq n/2$. Also, since $|x_i| \leq 0.1/(Q\sqrt{n})$, Chernoff bounds give:

$$\mathbf{Pr}\left[\left|\sum_{i>1}x_iv_i\right| \geq t\right] \leq 2e^{-50(Qt)^2}$$

Therefore, with probability at least $1 - 1/(nQ)^3$, $|\sum_{i>1}v_iX_i| \leq \sqrt{\log(nQ)}/Q$.

If $Qrn \geq \Omega(\sqrt{\log(nQ)})$, then we have

$$-Q\sum_{i>1}v_iX_i - \frac{1}{2}Qr\sum_{i>1}X_i^2 \leq -\Omega(\sqrt{\log(nQ)})$$

We conclude that the probability of descent is upper bounded by $\mathbf{Pr}\left[v_1X_1 \leq -\Omega(\sqrt{\log(nQ)})\right]$. This probability is exactly $\Phi(-l)$, where $\Phi$ is the cumulative density of a standard normal and $l = \Omega(\sqrt{\log(nQ)})$. By a naive upper bound, we see that

$$\Phi(-l) = \frac{1}{\sqrt{2\pi}}\int_l^\infty e^{-x^2/2}\,dx$$
$$\leq \frac{C}{l}\int_l^\infty xe^{-x^2/2}\,dx$$
$$= \frac{C}{l}e^{-l^2/2}$$

Since $l = \Omega(\sqrt{\log(nQ)})$, we conclude that with probability at least $1 - 1/poly(nQ)$, we have $f(y) - f(x^*) \geq f(x) - f(x^*)$.

Otherwise, we are in the case that $Qrn \leq O(\sqrt{\log(nQ)})$. Arguing simiarly as before, with high probability, our objective function and each coordinate can change by at most $O(\sqrt{\log(nQ)}/(Qn))$.

Next, we extend our proof to any symmetric distribution $\mathcal{D}$. Since $\mathcal{D}$ is rotationally symmetric, if we parametrize $v = (r, \theta)$ is polar-coordinates, then the p.d.f. of any scaling of $\mathcal{D}$ must take the form $p(v) = p_r(r)u(\theta)$, where $u(\theta)$ induces the uniform distribution over the unit sphere. Therefore, if $Y$ is a random variable that follows $\mathcal{D}$, then we may write $Y = Rv/\|v\|$, where $R$ is a random scalar with p.d.f $p_r(r)$ and $v$ is a standard Gaussian vector and $R, X$ are independent.

As previously argued, $\|v\| \in [0.5n, 1.5n]$ with exponentially high probability. Therefore, if $R \geq \Omega(\sqrt{\log(nQ)}/Q)$, the same arguments will imply that $Y$ is a descent direction with polynomially small probability. Thus, when $Y$ is a descent direction, it must be that $R \leq \Omega(\sqrt{\log(nQ)}/Q)$ and as argued previously, our lower bound follows similarly.

$\square$

## B    PROOFS OF SECTION 4

*Proof of Theorem 13.* By the Gaussian version of Theorem 7 (full rank version of Theorem 10), as long as our binary search sweeps between minimum search radius $r \leq \frac{3}{5Q\sqrt{n}}\|x - x^*\|$ and

maximum search radius of the diameter of the whole space $R = \|\mathcal{X}\|$, the objective value will decrease multiplicatively by $1 - \frac{1}{5nQ}$ in each iteration with constant probability. Therefore, if $\|x_t - x^*\| \geq 2Q\epsilon$ and we set $r = \frac{\epsilon}{\sqrt{n}}$ and $R = \|\mathcal{X}\|$, then with high probability, we expect $f(x_T) - f(x^*) \leq \beta Q^2 \epsilon^2$ after $T = O(nQ \log(\|\mathcal{X}\|/(Q\epsilon)))$ iterations, where we note that $F = f(x_0) - f(x^*) \leq \beta \|\mathcal{X}\|^2$ by smoothness.

Otherwise, if there exists some $x_t$ such that $\|x_t - x^*\| \leq 2Q\epsilon$, then $f(x_T) - f(x^*) \leq f(x_t) - f(x^*) \leq 4\beta Q^2 \epsilon^2$. Therefore, by strong convexity, we conclude that in either case, $\|x_T - x^*\| \leq 2Q^{3/2}\epsilon$. Finally note that each iteration uses a binary search that requires $O(\log(R/r)) = O(\log(n\|\mathcal{X}\|/\epsilon))$ function evaluations.

Therefore, by combining these bounds, we derive our result. The low-rank result follows from applying Theorem 10 and Theorem 11 instead. $\qquad \square$

*Proof of Theorem 14.* Let $H = O(nQ \log(Q))$ be the number of iterations between successive radius halving and we initialize $R = \|\mathcal{X}\|$ and half $R$ every $H$ iterations. We call the iterations between two halving steps an epoch. We claim that $\|x_i - x_0\| \leq R$ for all iterations and proceed with induction on the epoch number. The base case is trivial.

Assume that $\|x_i - x_0\| \leq R$ for all iterations in the previous epoch and let iteration $i_s$ be the start of the epoch and iteration $i_s + H$ be the end of the epoch. Then, since $\|x_{i_s} - x^*\| \leq R$, we see that $f(x_{i_s}) - f(x^*) \leq \beta R^2$ by smoothness. If $\frac{R}{4\sqrt{Q}} \leq \|x_i - x^*\| \leq 4\sqrt{Q}R$ for all $i$ in the previous epoch, then by the Gaussian version of Theorem 7 (Theorem 10), since we do a binary sweep from $\frac{R}{4Q}$ to $4\sqrt{Q}R$, we can choose $\mathcal{D}$ accordingly so that we are guaranteed that our objective value will decrease multiplicatively by $1 - \frac{1}{5nQ}$ with constant probability at a cost of $O(\log(Q))$ function evaluations per iteration. This implies that with high probability, after $O(nQ \log(Q))$ iterations, we conclude

$$f(x_{i_s+H}) - f(x^*) \leq \frac{1}{4Q}(f(x_{i_s}) - f(x^*)) \leq \frac{\alpha}{4}\|x_{i_s} - x^*\|^2 \leq \frac{\alpha}{4}R^2$$

Otherwise, there exists some $1 \leq j \leq H$ such that $\|x_{i_s+j} - x^*\| \geq 4\sqrt{Q}R$ or $\|x_{i_s+j} - x^*\| \leq \frac{R}{4\sqrt{Q}}$. If it is the former, then by strong convexity, $f(x_{i_s+j}) - f(x^*) \geq \alpha\|x_{i_s+j} - x^*\|^2 \geq 2\beta R^2$, which contradicts the fact that $f(x_{i_s}) - f(x^*) \leq \beta R^2$ by smoothness. If it is the latter, then by smoothness, we reach the same conclusion:

$$f(x_{i_s+H}) - f(x^*) \leq f(x_{i_s+j}) - f(x^*) \leq \beta\|x_{i_s+j} - x^*\|^2 \leq \frac{\alpha}{4}R^2$$

Therefore, by strong convexity, we have

$$\|x_{i_s+H} - x^*\| \leq \sqrt{\frac{f(x_{i_s+H}) - f(x^*)}{\alpha}} \leq \frac{R}{2}$$

And our induction is complete. Therefore, we conclude that after $\log(\|\mathcal{X}\|/\epsilon)$ epochs, we have $\|x_T - x^*\| \leq \epsilon$. Each epoch has $H$ iterations, each with $O(\log(Q))$ function evaluations and so our result follows.

The low-rank result follows from applying Theorem 10 and Theorem 11 instead. However, note that since we do not know the latent dimension $k$, we must extend the binary search to incur an extra $\log(n)$ factor in the binary search cost. $\qquad \square$

# C FIGURES

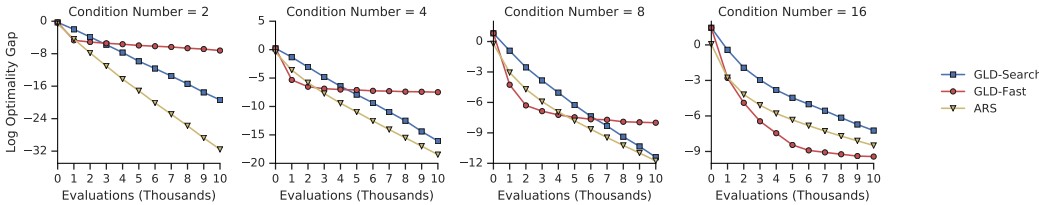

Figure 2: The average optimality gap by the condition number of the objective function.

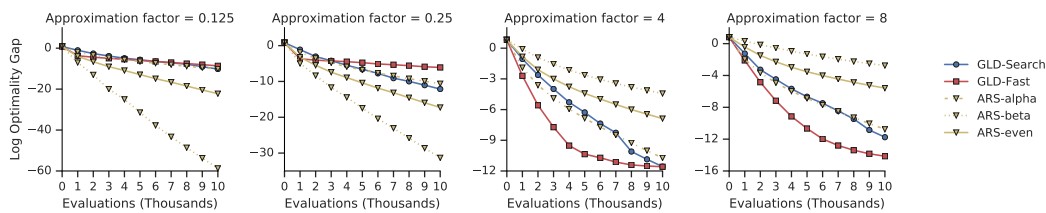

Figure 3: The average optimality gap by the accuracy of the condition number estimate, where approximation factor is the ratio of estimated to true condition number. The dimension $n = 20$.

## C.1 BBOB FUNCTION PLOTS

Figure 4: Convergence plot for the BBOB Rastrigin Function.

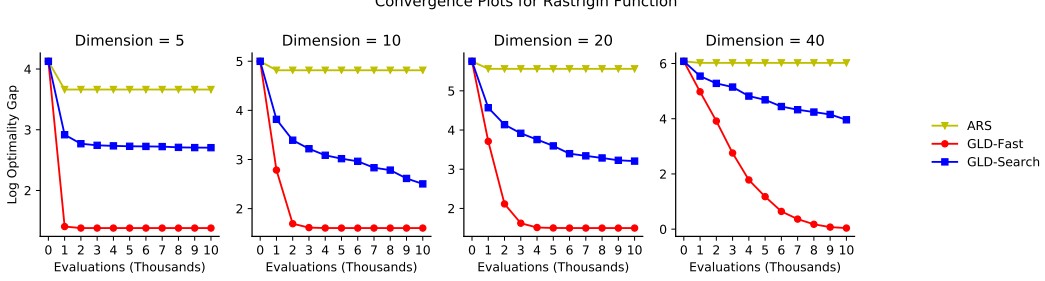

Figure 5: Convergence plot for the BBOB BentCigar Function.

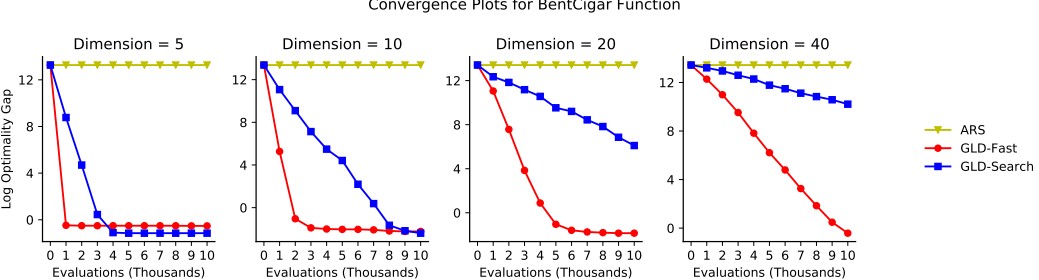

Figure 6: Convergence plot for the BBOB BuecheRastrigin Function.

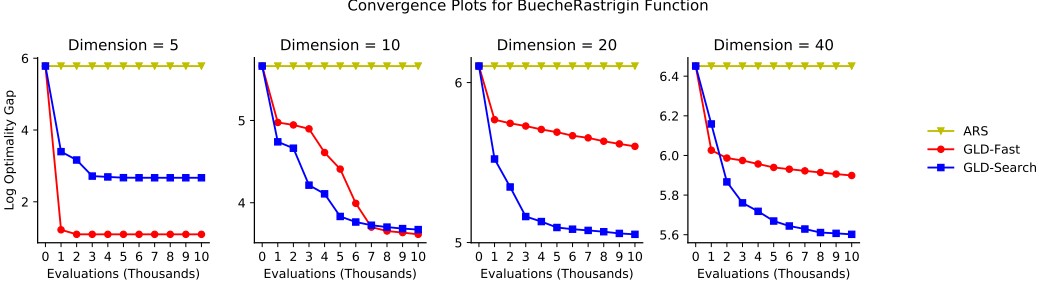

Figure 7: Convergence plot for the BBOB DifferentPowers Function.

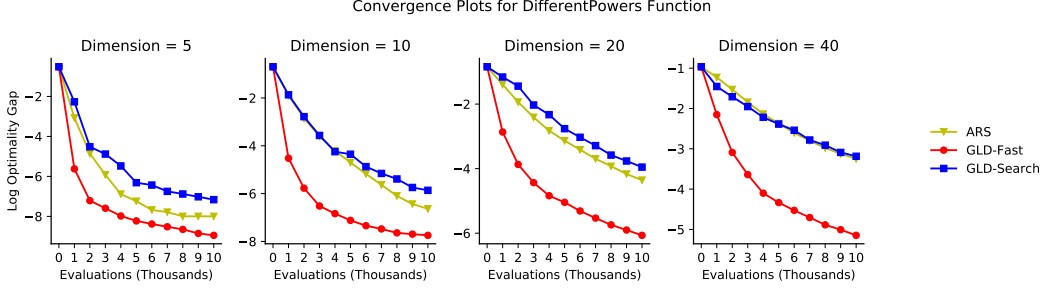

Figure 8: Convergence plot for the BBOB Discus Function.

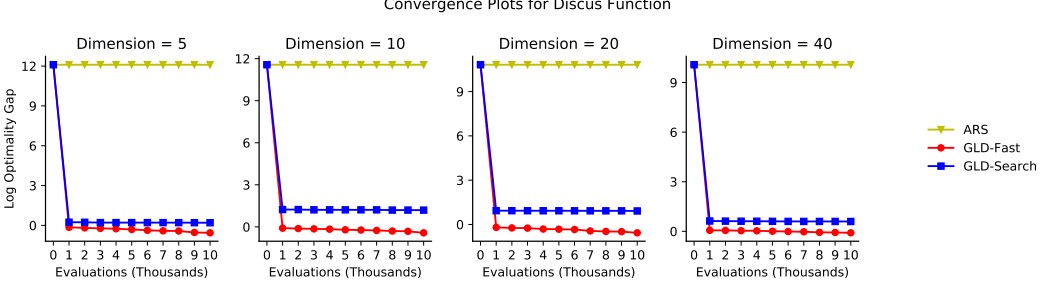

Figure 9: Convergence plot for the BBOB Ellipsoidal Function.

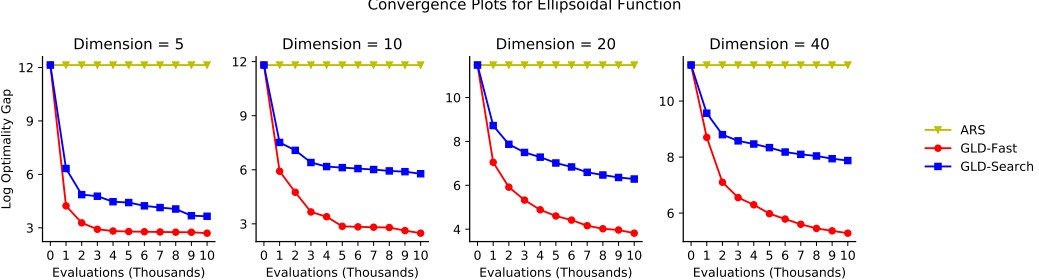

Figure 10: Convergence plot for the BBOB Katsuura Function.

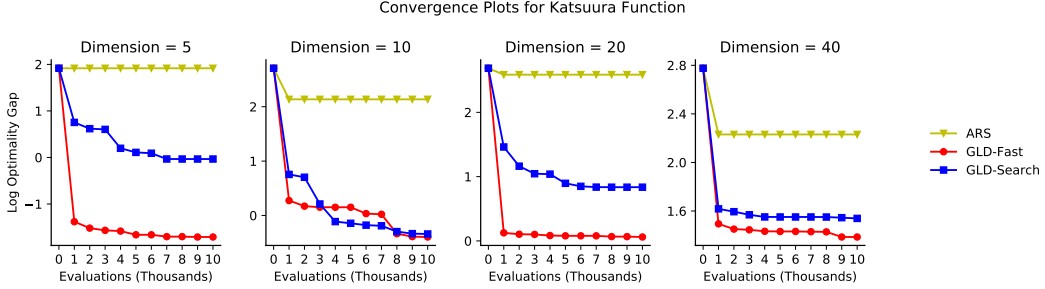

Figure 11: Convergence plot for the BBOB SchaffersF7 Function.

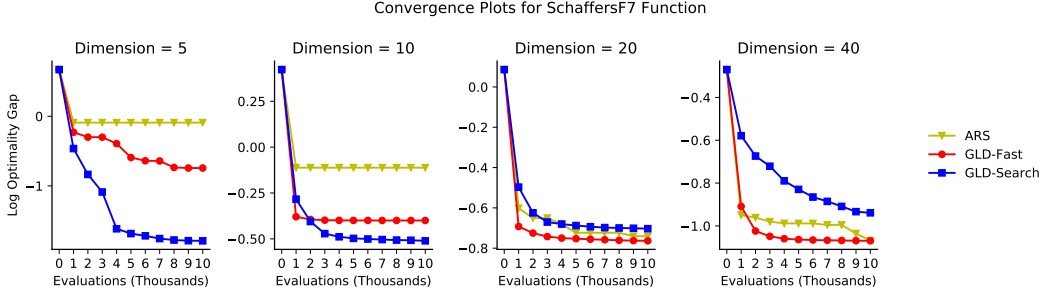

Figure 12: Convergence plot for the BBOB Ill-Conditioned SchaffersF7 Function.

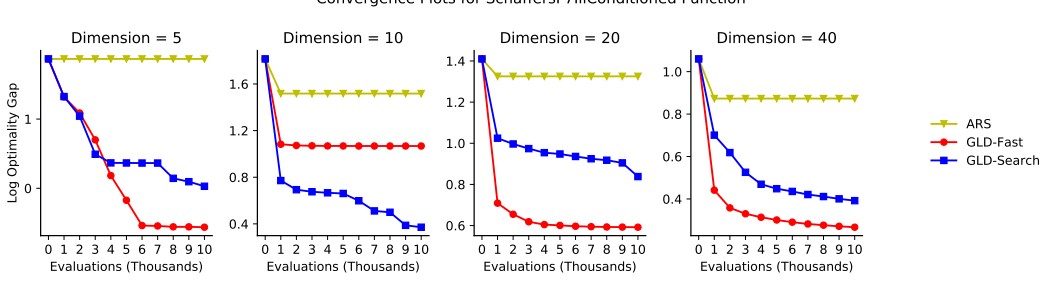

Figure 13: Convergence plot for the BBOB SharpRidge Function.

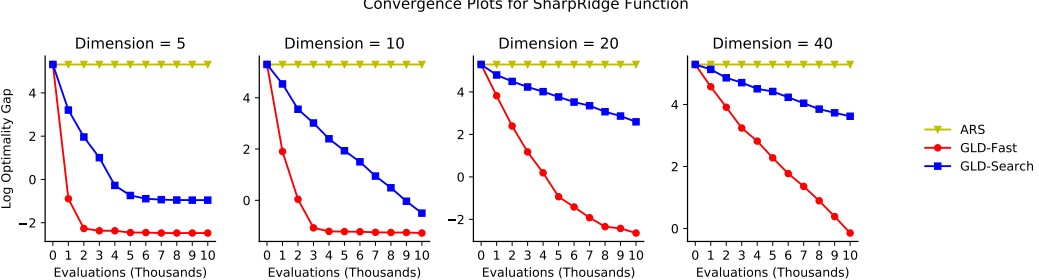

Figure 14: Convergence plot for the BBOB Weierstass Function.

Convergence Plots for Weierstass Function

## C.2   MUJOCO CONTROL PLOTS

Figure 15: Plot of maximum reward so far found by algorithm. Main line is the median trajectory across 3 runs.

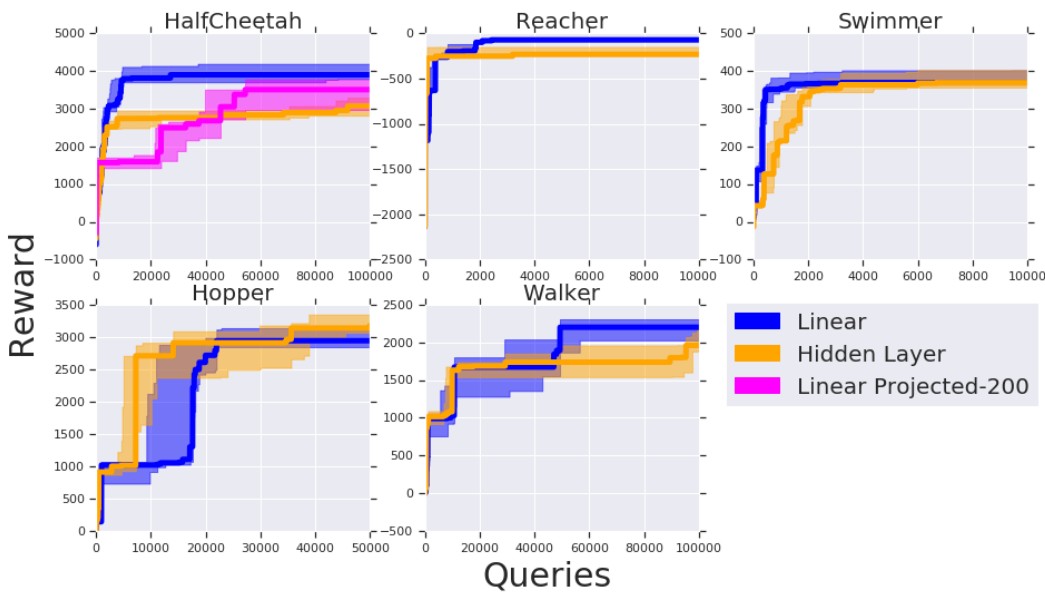

Figure 16: Example of rewards found by all samples by algorithm.

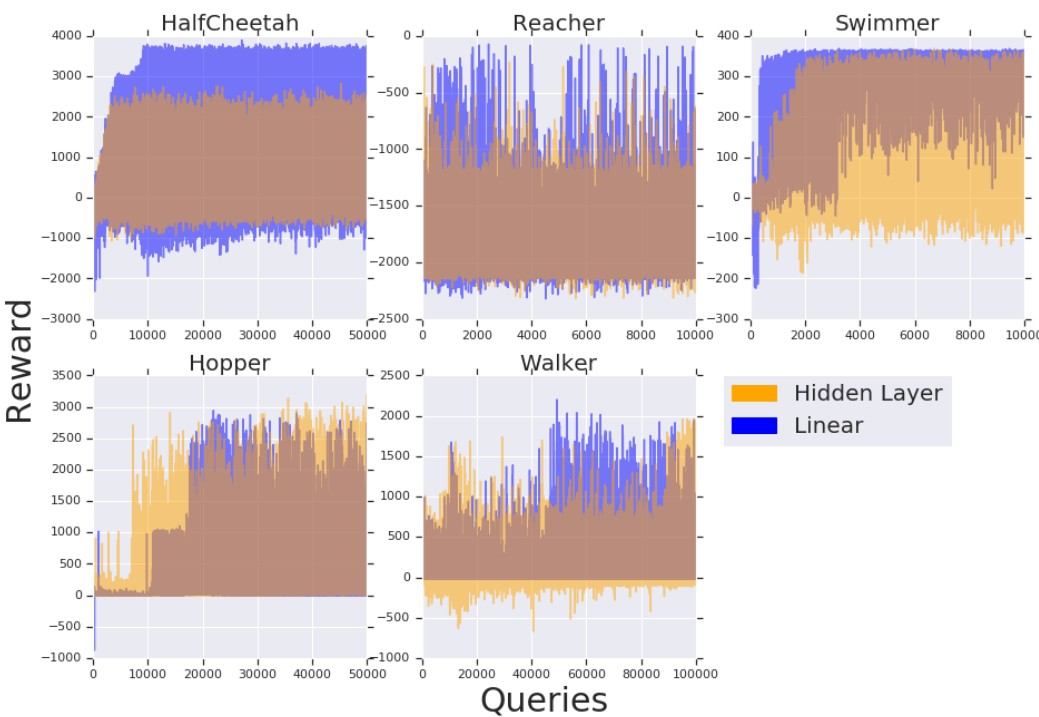

