# OpenReview forum: "Gradientless Descent: High-Dimensional Zeroth-Order Optimization"
_ICLR.cc/2020/Conference — Accept (Spotlight)_

### Official Review · AnonReviewer2 · 2019-10-22
**Official Blind Review #2**

**Rating:** 6

**Review:**

This paper proposes stable GradientLess Descent (GLD) algorithms that do not rely on gradient estimate. Based on the low-rank assumption on P_A, the iteration complexity is poly-logarithmically dependent on dimensionality. The theoretical analysis of the main results is based on a geometric perspective, which is interesting. The experimental results on synthetic and MuJoCo datasets validate the effectiveness of the proposed algorithms.

The theoretical contribution of this paper is nice and valuable. My main concern is the structure f(x) = g(P_A x) + h(x) looks somewhat limited. A more natural form is moving the perturbation into g, i.e, f(x) = g(P_A x + h(x)).

The experiments on Mujoco do not satisfy the assumption previous. Is there any real-world application which matches the theoretical analysis?

In summary, I think this is a good paper and tend to accept it.


**Experience Assessment:**

I have read many papers in this area.

**Review Assessment: Checking Correctness Of Derivations And Theory:**

I assessed the sensibility of the derivations and theory.

**Review Assessment: Checking Correctness Of Experiments:**

I assessed the sensibility of the experiments.

**Review Assessment: Thoroughness In Paper Reading:**

I read the paper at least twice and used my best judgement in assessing the paper.

---

> ### Author Response · Authors · 2019-11-13
> **Author Response to Official Blind Review #2**
>
> Thank you for your positive review.
>
> >> New Noise Model
> We thank the reviewer for the proposal of a new noise model. We would like to note the noise model proposed can be reduced to our current noise model. Note that using h_2(x) = g(P_Ax + h(x)) - g(P_Ax) reduces to our case with h_2(x) as our perturbation (i.e., h(x)). Furthermore, since we are working with the case when h(x) is small and g is smooth, we also know that h_2(x), by Taylor expansion, is approximately g’(P_Ax) h(x), which can be bounded. Therefore, any theoretical bounds we have can be translated to the proposed noise model.
>
> >> “The experiments on Mujoco do not satisfy the assumption previous. Is there any real-world application which matches the theoretical analysis?”
> Please see our meta-response.

---

### Official Review · AnonReviewer3 · 2019-10-23
**Official Blind Review #3**

**Rating:** 8

**Review:**

** Summary
The paper proposes a novel zeroth-order algorithm for high-dimensional optimization. In particular, the algorithm as an instance of direct search algorithms where no attempt is made to estimate the gradient of the function during the optimization process. The authors study the optimization of monotone transformations of strongly-convex and smooth functions and they prove complexity bounds as a function of the condition number, the dimensionality and the desired accuracy. These results are also extended to the case where the function actually depends on a lower-dimensional input. Without any knowledge of the actual subspace of interest, the algorithm is able to adapt to the (lower) dimensionality of the problem. The proposed algorithms are tested on synthetic optimization problems and in a few Mujoco environments for policy optimization.

** Overall evaluation
The paper is a solid theoretical and algorithmic contribution to the zeroth-gradient optimization literature. The positive aspects of the paper are:
- Novel algorithm with strong theoretical guarantees improving or generalizing previous state-of-the-art methods.
- Ability to adapt to low-dimensional problems and more in general to monotone transformations of convex functions.
- Efficient version.

Negative aspects of the paper that the authors may address are:
- The empirical validation is rather weak at the moment. It provides some evidence of the effectiveness of the proposed method but it uses only one baseline and a very few type of optimization problems. Although in my opinion the main contribution is on the theoretical side, a more thorough empirical validation would be welcome.
- Some theorem statements can be made clearer and some comparisons should be more explicit (see detailed comments later).

Detailed comments:
1- The authors explicitly mentioned in the introduction that they do not compare/discuss alternative approaches such as Bayesian optimization (BO). Although I agree the approaches may be different, BO is probably the most popular type of black-box optimization. Furthermore, many methods (e.g., GP-UCB https://arxiv.org/abs/0912.3995) come with strong theoretical guarantees on the regret and so optimization performance both under the Bayesian assumption (i.e., the function is generated from a prior) and the "frequentist" case (i.e., the function is an arbitrary element of a bounded RKHS). Furthermore, there are also adaptive BO methods that adapt to the actual dimensionality of the problem, in a similar spirit as the low-dimensional case studied in this paper. See e.g., https://arxiv.org/abs/1903.05594 and http://papers.nips.cc/paper/8115-efficient-high-dimensional-bayesian-optimization-with-additivity-and-quadrature-fourier-features. I would appreciate if the authors would at least provide a high-level discussion on similarities and differences between these type of approaches.
2- Thm7: r = 2^k1 C_1 and r = 2^-k2 C_2 are the only two possible radii? Is the statement valid for any choice in the range?
3- Thm13: Unlike the statements in Sect.3.2 and 3.3, here the result is reported in terms of x_T (instead of f(x_T)). This is perfectly fine, but the guarantee you obtain is not an epsilon accuracy, but Q^{3/2}epsilon. If we want to obtain an epsilon accuracy, how much is the number of evaluation going to change? It seems like it would just make an additional Q appear in the log, but I would like the authors to confirm.
4- Thm13: "High-probability": could you make this more explicit? Can I make the probability arbitrarily close to 1? How would it appear in the number of iterations? As just a log(1/delta) term?
5- Thm14 is reported for "suitable parameters". Although this choice of parameter actually appears in Alg.2, it would be more complete to report it in the statement as well.
6- Fig1 bottom line, first two charts display a weird behavior for GLD-Fast, where the error seems to plateau and spot decreasing. Can you explain why this is happening? Is it due to wrong parameters \hat alpha and \hat beta?

Minor comments:
- In the proof of Lem.8, it would be helpful to have a graphical representation of the spheres and the hyperspherial caps.
- In the proof of Lem.15, you mention "strong smoothness assumption", it should be just smoothness.
- It would be helpful to have more intuition on the why the algorithm is able to adapt to the actual dimensionality of the problem. My understanding is that the probability to pick a point of lower value is increased and since the algorithm is testing different radii and pick the best point, it successfully adapt to this better situation.

**Experience Assessment:**

I have published one or two papers in this area.

**Review Assessment: Checking Correctness Of Derivations And Theory:**

I assessed the sensibility of the derivations and theory.

**Review Assessment: Checking Correctness Of Experiments:**

I assessed the sensibility of the experiments.

**Review Assessment: Thoroughness In Paper Reading:**

I read the paper at least twice and used my best judgement in assessing the paper.

---

> ### Author Response · Authors · 2019-11-13
> **Author Response to Official Blind Review #3**
>
> Thank you for your review and encouraging comments!
>
> >> Empirical Evaluation
> Please see our meta-response.
>
> >> 1) Comparison to Bayesian Optimization
> Please see our meta-response.
>
> >> 2) “Thm7: r = 2^k1 C_1 and r = 2^-k2 C_2 are the only two possible radii? Is the statement valid for any choice in the range?”
> We would like to clarify that our theorem proves the existence of two radii that can be found via binary search that satisfy our descent criterion. The reviewer is correct that the statement is valid for any choice in the range. We would like to note that we only have to show the existence of two radii for our algorithms to provably succeed.
>
> >> 3) “Thm13: Unlike the statements in Sect.3.2 and 3.3, here the result is reported in terms of x_T (instead of f(x_T)). This is perfectly fine, but the guarantee you obtain is not an epsilon accuracy, but Q^{3/2}epsilon. If we want to obtain an epsilon accuracy, how much is the number of evaluation going to change? It seems like it would just make an additional Q appear in the log, but I would like the authors to confirm.”
> We would like to clarify that our theorem will sometimes return an accuracy that is dependent on Q in that case where an upper bound on Q is unknown. The reviewer is correct that the iteration complexity will simply make an addition Q appear in the logarithm.
>
> >> 4) “Thm13: "High-probability": could you make this more explicit? Can I make the probability arbitrarily close to 1? How would it appear in the number of iterations? As just a log(1/delta) term?”
> We would like to clarify that high probability means that the failure probability is 1/poly(n) and can be made arbitrarily close to 1 (as long as the failure probability is not exponentially small). If a dependence on failure probability delta were to be made explicit, the reviewer is correct that it will be an additional log(1/delta) (and remove a log(n) from the iteration bound).
>
> >> 5) “Thm14 is reported for ‘suitable parameters’. Although this choice of parameter actually appears in Alg.2, it would be more complete to report it in the statement as well.”
> We would like to clarify that we try to only include parameters that appear in the final iteration complexity bound, so as to not overwhelm the reader. The reviewer is correct in stating that all details are mentioned in the final algorithm.
>
> >> 6) “Fig1 bottom line, first two charts display a weird behavior for GLD-Fast, where the error seems to plateau and spot decreasing. Can you explain why this is happening? Is it due to wrong parameters \hat alpha and \hat beta?”
> We would like to clarify that GLD-Fast can often overshoot when it is extremely close to the optima due to the restrictions on the radii sizes and the binary search procedure can try a lot of large radii sizes, leading to overshooting. We would like to note that GLD-Fast only plateaus when the optimality gap is exceedingly small.
>
> >> Minor Comments.
> We thank the reviewer for the minor comments and we have edited the paper to get rid of typos and clarify things.

---

### Official Review · AnonReviewer1 · 2019-10-29
**Official Blind Review #1**

**Rating:** 6

**Review:**


Update after rebuttal: I found the rebuttal convincing and I liked the fact that concerns regarding empirical justification were addressed. Consequently, I increase my score from "Weak Reject" to "Weak Accept".
--------------------------
This paper focuses on derivative-free, or zero-th order, optimization. That is the setting where a function may be continuous and/or smooth and/or (quasi)-convex, however, we do not have access to the gradients. As such, it is not possible to apply standard gradient descent methods. The paper proposes an algorithm for "gradientless" descend. The basic intuition is that by careful randomly sampling it is quite probable that a lower objective value will be attained. Doing so recursively can then lead to the optimum with high probability. Clearly, in such a setting it is quite important to clarify what is "careful random sampling". To this end the paper casts this as sampling from a Gaussian ball of specific radius, chosen such that the samples are with high probality below the current level set (the hyperplance of equivalent solutions f(x) as our current solution f(x_t)). The paper derives and proves various theorems on how to select the optimal radius and how to perform the sampling. Specifically, the case of strongly convex and smooth functions is analyzed, however, the paper also shows how this generalizes to functions after a monotone transformation (thus leading to quasi-convex functions) and with extra error perturbations. The proposed algorithm is compared on a synthetic experiment, and a selection of MuJoCo benchmarks.

Strengths:
+ The derivations and the theorems are non-trivial. There is some serious analysis regarding the selection of radius of Gaussian balls. I would like to congratulate the authors for this. I particularly like the extension to having a perturbing function h(x), leading to a more realistic setup.
+ I also particularly like that the algorithm is able to recover subspaces automatically. This definitely makes the algorithm much more practical and more efficient.
+ The writing and the presentation are rather clear and well taken care of. Although there are several theorems, it was not too hard to follow the flow of the paper. The algorithm boxes are also concise and clear, helping with understanding the final result.

Weaknesses:
+ Although the contributions of the work are mostly on the theoretical side, I have a hard time grasping how useful is the algorithm in practice. For one, there is the assumption of strongly convex and smooth function. Granted, there is the relaxed case of having the perturbing function h(x), howevrer, in that case it seems that the algorithm becomes a slower by an order of 60 δ k Q_g(A). How fast or slow is this in practice? Even with applying the monotone function, the algorithm becomes more practical by being applicable to quasi-convex setups. However, how realistic is that a function will in practice be strictly monotone? While most of this may be hard to be theoretically proven, they can be experimentally tested.

+ Also, given that the paper is interested in black box functions, we cannot have much information regarding the function. So, what happens when the function is not strongly convex or not always smooth? Furthermore, how realistic is to know the condition number, that is the maximum derivative (or an upper bound of it) since we do not have access to the gradients in the first place?

+ I would say that the paper could benefit from a more extensive experimental section. Currently only a single synthetic function is analyzed, also under a single monotone exponential transformation. From a more practical point of view, MuJoCo environments are also examined. However, there exist no comparisons with other methods in the literature, including ARS. Another relevant algorithm to compare with would be the stochastic tree points (Bergou et al., 2019), if not experimentally at least theoretically. In the end, it quite unclear whether the algorithm work well in practice. Some experiments that could shed light would relate to how sensitive the algorithm is to the convexity/smoothness assumptions, how sensitive the algorithm is to the perturbing function h(x), how sensitive is the algorithm to the present of a lower-dimensional subspace that needs to be discovered. And for the MuJoCo experiments, the algorithm can compare at least with ARS.

+ In the experiments it seems the paper is particularly good in high dimensions. Can this be more precisely connected to the derived theory in the discussion of the experiments? Does this relate to the better subspaces k that are discovered automatically by the algorithm?

+ It is unclear how many evaluations are needed per step, that is what is the K value in the algorithm box? Also, there are at least two balls to sample from, so twice as many evaluations, correct?

+ It is unclear why Bayesian Optimization is not considered for at least comparing experimentally. Currently, the paper discards them on the grounds that they do not provide strong theoretically guarantees. However, it would be interesting to examine at least in practice how good/bad are these algorithms in comparison to the proposed one. Two recent bayesian optimization papers that can be considered for continuous and discrete inputs are

[1] BOCK: Bayesian Optimization with Cylindrical Kernels, C. Oh, E. Gavves, M. Welling, ICML 2018
[2] BOCS: Bayesian Optimization of Combinatorial Structures, R. Baptista, M. Poloczek, ICML 2018

+ The paper does not have a conclusion. That shows great sloppiness. Also, i/n the abstract, do you mean to say k>=n or k<=n?

To conclude, I recommend weak rejection only because I am not completely convinced by the experiments and do not know if the proposed algorithm is competitive against reasonable baselines and in more complex setups. I am more than happy to upgrade my score if experiments become more clear.

**Experience Assessment:**

I have published one or two papers in this area.

**Review Assessment: Checking Correctness Of Derivations And Theory:**

I assessed the sensibility of the derivations and theory.

**Review Assessment: Checking Correctness Of Experiments:**

I carefully checked the experiments.

**Review Assessment: Thoroughness In Paper Reading:**

I read the paper at least twice and used my best judgement in assessing the paper.

---

> ### Author Response · Authors · 2019-11-13
> **Author Response to Official Blind Review #1**
>
> We thank Reviewer 1 for the compliments and suggestions.
>
> *** In response to the main concern, please see our meta-response, where we have added extra experimentation in Appendix C to compare GLD against Accelerated Random Search on numerous modified Blackbox functions. For Mujoco, we have also added comparisons against the reward threshold in the ARS (not to be confused with Accelerated Random Search) paper for RL [Mania18] to show that our algorithm remains competitive in finding the optimal reward. If after looking at the experiments and our other responses you believe that we have addressed your concerns well and our paper is interesting enough to be accepted at ICLR, we hope that you would kindly increase your rating.
>
> We address other concerns raised below.
>
> >> Fast or slow in practice?:
> We believe that our theory guides practice and (as commonplace in optimization literature) algorithms that provide guarantees only in the convex setting are often useful even when applied in a non-convex setting.  Also, please see our meta-response.
>
> >>Relaxed case?
> We would like to clarify that the perturbation function h(x) is not a noise term but any small adversarial addition. We would like to note that this does not slow down the convergence of the algorithm until the error becomes smaller than δ k Q_g(A). Furthermore, we demonstrate in our benchmarks that our algorithms do well in high dimensions (often much better than others), even for functions that are not explicitly low rank. This shows that our algorithm is practically quite effective in this regime. We believe that this is explained via our convergence bounds when the function is dominant only in a few directions.
>
> >> Strict monotoncity?:
> We would like to note that our algorithm does not assume monotone functions, rather monotone transformations--this follows from the fact that we depend only on the ranks of function values. In practice, monotone transformations (such as applying a log transformation) are often applied prior to optimization.
>
> >> “So, what happens when the function is not strongly convex or not always smooth? Furthermore, how realistic is to know the condition number, that is the maximum derivative (or an upper bound of it) since we do not have access to the gradients in the first place?”
> This is a valid point: condition numbers are not known in practice and convexity is mostly non-existent. Our GLD-Search algorithm copes with this by applying an extensive binary search procedure. However, as experimentally verified on MuJoCo and newly added BBOB benchmarks, we see that performance does not deteriorate significantly due to lack of knowledge of condition numbers or lack of convexity. More is mentioned in our general response.
>
> >> “In the experiments it seems the paper is particularly good in high dimensions. Can this be more precisely connected to the derived theory in the discussion of the experiments? Does this relate to the better subspaces k that are discovered automatically by the algorithm?”
> In the Mujoco benchmarks, we would like to note that we explicitly tested affine invariance by using a lift of our input space from 17 variables to 200 variables. We discuss our results in the experiments section that GLD is not sensitive to a massive increase in dimensionality. This does indeed relate to the subspaces of lower dimension (k = 17 in this case) that are automatically discovered by the algorithm.
>
> >> “It is unclear how many evaluations are needed per step, that is what is the K value in the algorithm box? Also, there are at least two balls to sample from, so twice as many evaluations, correct?”
> We would like to clarify that the K value is the number of evaluations per iteration and is set at the beginning of both GLD algorithms. This is required since ball sampling is sensitive to the radius and so K radii is attempted, via binary search. There is only one ball that is sampled at a specific time in the algorithm. While the proofs use a two-ball intersection analysis, please note that this is purely for analysis. Our search over ball radii will increase the number of evaluations, but this is accounted for in the theorem and proofs.
>
> >> Bayesian Optimization Comparisons
> Please see our meta response.
>
> >> Conclusion and “Also, i/n the abstract, do you mean to say k>=n or k<=n?”
> We thank the reviewer for the suggestion and for catching the typo. We have changed the abstract and have added a conclusion. The conclusion now briefly summarizes our contributions and provides possibilities for new research directions.
>
> ########################## References ##########################
> [Mania18]: Simple random search provides a competitive approach to reinforcement learning, NeurIPS 2018.

---

### Author Response · Authors · 2019-11-13
**Meta-Response**

We thank the reviewers for their time and extensive feedback, which is helpful for improving the paper. We have provided a revised version of the paper with more extensive experimentation and clarifications.

Below, we provide answers to common questions:

-----------------1. Empirical evaluation - Please see Section C.1 of the Appendix. In line with suggestions and concerns, we have **added numerous experimentation with BBOB (BlackBox Optimization Benchmarking) functions** using GLD and Accelerated Random Search.

This further demonstrates the utility of Gradientless Descent on functions that are not strongly smooth or convex and/or have multiple local optima.  GLD also seems to perform better in higher dimensions, implying that the algorithm is not sensitive to the lack of explicit low-dimensional structure. This result is not coincidental. GLD is invariant to the scaling of function values (invariance to monotonic transformations) and does a binary search over the step sizes. Accelerated Random Search is more reliant on the function’s smoothness, because it performs an adaptive gradient from estimated gradients.

We would like to first note that algorithms such as Adam [Reddi18] and Adagrad [Duchi10] only have guarantees in convex settings, but they are very successfully applied to non-convex settings. In non-convex settings, our theory guarantees the convergence to the “basin of attraction” within a locally strongly convex and smooth structure (and more generally to monotone transformations of such structures). In our original submission, we showed empirically that our algorithm is successfully applied to MuJoCo benchmarks, where the objective function is not strongly convex and smooth.


-----------------2. Bayesian Optimization (BO) Comparisons - We agree that Bayesian optimization is a good candidate, often used to solve these problems. In light of these concerns, we have added some comparisons in our introduction. However, they are not within the context of our paper and our comparisons for multiple reasons:

**** While there are known regret bounds for Bayesian optimization on non-convex problems, they are considerably narrower than bounds available under the assumption of convex objectives [Freitas12].

**** BO is a complex system of algorithms, which consists of (a) a model-building algorithm (e.g. Gaussian Process), which often includes hyperparameter optimization and regularization, (b) an acquisition function and an optimization strategy (which can be quite complicated [Wilson18]) for the acquisition function, and sometimes (c) pre-processing for the support points. Given all this complexity, there is room for both misconfiguration of a BO system, and over-adaption of a BO system to a given data set, and a proper comparison is outside the time and space of our paper. We further note that because GLD is also practially used as a component of (b) in a large-scale industrial zero-order optimization system [Golovin17], our theoretical analysis also benefits understanding of BO.

**** Bayesian optimization is notoriously slow for the inversion of the kernel matrix for the evaluation of the acquisition function [Rasmussen06], as shown in Figure 6 of [Golovin17].  The cost of generating a new step in a Bayesian Optimization system can be large, sometimes even dominating the cost of evaluating the objective function.  Consequently, while BO is valuable for expensive functions; there is an important niche where one needs to rapidly optimize functions that aren't expensive to compute.



########################## References ##########################
[Rasmussen06]: Gaussian Processes for Machine Learning, TextBook, 2006.
[Freitas12]: Exponential Regret Bounds for Gaussian Process Bandits with
Deterministic Observations, ICML 2012.
[Duchi11]: Adaptive Subgradient Methods for Online Learning and Stochastic Optimization, COLT 2010.
[Reddi18]: On the Convergence of Adam and Beyond, ICLR 2018.
[Wilson18]: Maximizing acquisition functions for Bayesian optimization, NeurIPS 2018.
[Golovin17]: Google Vizier: A Service for Black-Box Optimization, KDD 2017.

---

### Decision · Program_Chairs · 2019-12-19

**Decision:**

Accept (Spotlight)

**Comment:**

The paper considers an interesting algorithm on zeorth-order optimization and contains strong theory. All the reviewers agree to accept.